# The Prediction of Essential Medicines Demand: A Machine Learning Approach Using Consumption Data in Rwanda

**Francois Mbonyinshuti** [1,2,*] **, Joseph Nkurunziza** [3] **, Japhet Niyobuhungiro** [4] **and Egide Kayitare** [5]

1   African Center of Excellence in Data Science (ACEDS), College of Business and Economics, University of Rwanda, Kigali P.O. Box 4285, Rwanda
2   Human Resource for Health Secretariat, Ministry of Health, Kigali P.O. Box 84, Rwanda
3   College of Business and Economics, University of Rwanda, Kigali P.O. Box 4285, Rwanda; nkurunzizaj@gmail.com
4   National Council for Science and Technology (NCST), Kigali P.O. Box 2285, Rwanda; japhetniyo@gmail.com
5   College of Medicine and Health Sciences, University of Rwanda, Kigali P.O. Box 4285, Rwanda; egide.kayitare@gmail.com
*   Correspondence: francopha78@gmail.com or francois.mbonyinshuti@moh.gov.rw; Tel.: +25-07-8859-8986

**Abstract:** Today's global business trends are causing a significant and complex data revolution in the healthcare industry, culminating in the use of artificial intelligence and predictive modeling to improve health outcomes and performance. The dataset, which was referred to is based on consumption data from 2015 to 2019, included approximately 500 goods. Based on a series of data pre-processing activities, the top ten (10) essential medicines most used were chosen, namely cotrimoxazole 480 mg, amoxicillin 250 mg, paracetamol 500 mg, oral rehydration salts (O.R.S) sachet 20.5 g, chlorpheniramine 4 mg, nevirapine 200 mg, aminophylline 100 mg, artemether 20 mg + lumefantrine (AL) 120 mg, Cromoglycate ophthalmic. Our study concentrated on the application of machine learning (ML) to forecast future trends in the demand for essential drugs in Rwanda. The following models were created and applied: linear regression, artificial neural network, and random forest. The random forest was able to predict 10 selected medicines with an accuracy of 88 percent with the train set and 76 percent with the test set, and it can thus be used to forecast future demand based on past consumption data by inputting a month, year, district, and medicine name. According to our findings, the random Forest model performed well as a forecasting model for the demand for essential medicines. Finally, data-driven predictive modeling with machine learning (ML) could become the cornerstone of health supply chain planning and operational management.

**Keywords:** forecasting models; essential medicines; consumption data; health supply chain; machine learning; Rwanda

## 1. Introduction

Predicting future trends provides additional value for improved healthcare system management in today's global business trend and step forward technologies [1]. After all, the healthcare system is going to undergo a huge data revolution, with Artificial Intelligence (AI), predictive analytics, and business intelligence ready to increase efficiency and enhance health outcomes [2]. AI, especially ML, is expected to be widely employed to predict previously unknown patterns in disease, treatment, and care by 2030, according to the World Economic Forum [3]. Business growth in any sector as it is for healthcare, is all about adapting and pivoting to stay relevant in the market. You can't expect to be in business if your consumer needs change, but your supply chain doesn't. Even though the health supply chain is rapidly evolving, predictive models can help maintain a high level of performance by providing insights into patterns based on historical data.

Medicines quantification is the process of calculating the quantities and prices of commodities necessary for a certain health program (or facility) and determining when the medical item should be delivered to guarantee that the intended program or service is

continually given to the intended users [4]. For reliable and successful quantification of essential medicines, information about consumption data, procurement period, prescriptions, minimum and maximum inventory levels, inventory rotation, morbidity data, along with compliance with the use of an essential medicines list (EML) is required [5,6]. This high complex process entails predicting the number of essential medicines required and serves as a basis for determining the appropriate quantity to procure.

There are substantial barriers to developing and enhancing the efficiency of a well-organized supply chain of critical medications due to their high complexity, which is increasingly adding to people' delayed access to essential medicines [7]. Furthermore, there is no room for inaccuracy when it comes to an important and functioning health supply chain that assures the availability of essential medicines since it might have a harmful impact on people's health, socio-economic position, and day-to-day activities. Supply uncertainty for essential pharmaceuticals and life-saving products can be reduced with accurate demand forecasts [8,9].

In light of available resources, supply chain information and inventory levels, the required quantities of essential medicines should always be determined using an appropriate approach, such as prediction models, economic order quantity, or the Min/Max formula [10]. It's fair-minded to assume that collaborative forecasts based on end-user consumption data will help upstream supply chain managers minimize prediction error when employing forecasting models in the management of essential medicine supply and inventory control [11].

Whereas Rwanda has guaranteed that its citizens have access to quality healthcare and affordable medicines, the goal of this study is to see whether ML techniques might be used to improve demand forecasting accuracy and therefore optimize the availability of essential medicines [12]. Linear regression, artificial neural networks (ANN), and Random Forest have all been studied, suggested, and used in supply chain fields, with amazing outcomes on a range of difficulties [13,14]. Therefore, it is potentially relevant to undertake the benefit of ML application in health supply chain for predicting the future consumption of essential medicines in Rwanda. The aim of this study is to apply a ML approach to predict future trends for essential medicines consumption in Rwanda by using historical data extracted from the electronic logistic management information system (eLMIS), which is an electronic tool used in the management of health commodities in Rwanda, for a period of five years (2015–2019).

The architecture of this study is as follows: this portion serves as an introduction, and the third section discusses the nature of data and applied methodology. The data exploration, analysis and results were covered in the third section, and the explanation of experimental outcomes was covered in the fourth. The fifth section followed with conclusions and recommendations.

## 2. Materials and Methods

### 2.1. Settings

Rwanda is dedicated to increasing investment in universal healthcare. Rwanda has ensured that its residents have access to primary health care so far, with the community-based health insurance (CBHI) scheme covering over 84 percent of Rwandans. The country currently has a well-functioning, decentralized public health-care system in place, as well as a growing private health-care sector [15]. According to the Rwanda health supply chain structure, all public health facilities report monthly on the consumption of essential medicines using the eLMIS, which connects the Ministry of Health, Rwanda Medical Supply Ltd. (RMS), district-based RMS branches, and all public health facilities.

In Rwanda, all district-based public health institutions received the needed essential medicines and pharmaceutical supplies monthly from RMS Branches. Any purchase request from a health facility for important medications should be based on historical data, notably consumption and available stock. Under normal situations, essential pharmaceutical stock levels should not fall below the minimum and emergency stock level limitations

calculated based on past consumption data. The RMS, which is the national central medical stores, provide necessary essential medicines to district-based RMS branches, which may also cooperate with an existing faith-based medical store to assure medicines availability. If any essential medicines are unavailable, RMS Ltd. can procure them through private pharmaceutical wholesalers, such as BUFMAR (Bureau des Formations Médicales Agréées du Rwanda), a faith-based pharmaceutical and medical products supplier. In this perspective, local private pharmaceutical business companies are engaged in cases of emergency, but usually medicines are typically purchased from the international pharmaceutical market.

### 2.2. Sources, Types of Data and Data Pre-Processing

This study employed program data extrapolated from the eLMIS which is a digital tool used in management of health commodities in Rwanda. Data has been used and processed to be appropriately used in the prediction process. In this regard, one of the most essential goals of time series analysis is the prediction of future data, which may be utilized in the health supply chain [11]. While making strategic decisions under uncertainty, predictive analytics and models offer a great prospect. A univariate time series, in practice, is a collection of observations of a single random variable across period. In fact, the more the uncertainty in expectations, the higher the degree of inaccuracy in time series forecasting [16,17]. Subsequently, from different viewing platforms that can be applied in the forecast process, a suitable technique should be selected in consideration to the nature and amount of available historical data.

Data related to health supply chain management in Rwanda from 2015 to 2019 were used for this paper. They included consumption data, inventory data, ordering and re-ordering data, data on purchasing prices. They were extrapolated from the eLMIS under the authorization of the Ministry of Health through its department in charge of Clinical Services and Public Health. Although the eLMIS tool provided access to data at central level, district, and health facility levels, data were assessed at district level where all data from district-based facilities are compiled.

The read excel function from the panda's library was used to import data into DataFrame. Only five columns (variables) were retained to reduce the number of variables that are not useful to our study. They include the amount of drug consumed in quantity, the name of drug consumed, consumer districts, health center or hospital, year, and month of consumption. Data were collected over a five-year period, from 2015 to 2019, and they were available on either monthly or quarterly basis.

Data input errors and inaccuracies were cleaned up, such as none in a column of year and each character blocking data transformation to "Date_Time" type, and rows with none and each character were also eliminated. Since there were so many observations without districts, consumption at districts were predicted based on data from their health-care facilities. As the quantity had no decimals, the item and quantity of pharmaceutical purchased were converted to integers, and absolute functions were used to avoid any negative values.

The dataset had more than 500 drugs, but in our tasks, we only selected eight (8) most used drugs that are: Cotrimoxazole 480 mg tablet b/1000, amoxicillin 250 mg capsule b/1000, Paracetamol 500 mg tablet b/1000, O.R.S sachet 20.5 g b/100, Chlorpheniramine 4 mg tablet b/1000, Nevirapine 200 mg tablet b/60, aminophylline 100 mg tablet b/1000, arte 20 mg + Lumefantrine 120 mg tab (4 × 6) b/30, Cromoglycate disodic opht. solution 2%, Iodine Polyvidone 10% solution 200 mL b/1. According to the World Health Organization's Anatomical Therapeutic Chemical (ATC) classification system, the most consumed and selected which are considered in our study, are described in the Table 1:

**Table 1.** Description of selected essential medicines.

| ATC Code | Drug Name | Pharmacotherapeutic Group |
|---|---|---|
| J01EE01 | Cotrimoxazole (80 + 400) mg | Antibacterial for systemic use |
| J01CA04 | Amoxicillin 250 mg | Penicillin with extended spectrum |
| N02BE01 | Paracetamol 500 mg | Other analgesic and antipyretic, anilide |
| A07CA | Oral Rehydration Salt-Sachet 20.5 g | Low-osmolarity oral rehydration salts |
| R06AB04 | Chlorpheniramine 4 mg | Antihistamines for systemic use, substituted alkylamines |
| J05AG01 | Nevirapine 200 mg | Antivirals for systemic use, non-nucleoside reverse transcriptase inhibitors |
| R03DA55 | Aminophylline 100 mg | Ethylenediamine salt of theophylline |
| P01 BE52 | Artemether 20 mg + Lumefantrine 120 mg | Antimalarials, blood schizonticide |
| SO1GX01 | Cromoglycate Opht. 2% | Ophthalmological; other antiallergic |
| D11AC06 | Iodine Polyvidone 10% 200 mL | Dermatological preparation |

According to the Table 1, the selected essential medicines included a range of items such as antibacterial, analgesic, antihistamines, antiretroviral, antimalarials, ophthalmological and dermatological products. To reduce the vast number of few numbers in or to reduce over-fitting, data were processed before model training using log transformation methods. Quantity of the amount utilized were changed to float type. The modified numbers were returned to normal using exponential power, according to the prediction.

## 3. Data Exploration and Results

### 3.1. Description of Essential Medicines Consumption by Frequency

According to Table 2, the top three essential medicines that are most frequently consumed, by count, are Cotrimoxazole 480 Mg Tablet B/1000 (29301), Amoxicillin 250 mg Capsule B/1000 (24024), and Paracetamol 500 Mg Tablet B/1000 (23515). The Table 2 also indicated that the top two essential medicines that have been most consumed by the quantity of amount consumed are Cotrimoxazole 480 mg with the amount equal to 180,492,898, followed by Amoxicillin 250 MG with the amount equal to 179,323,559.

**Table 2.** Essential medicines consumed by frequency (2015–2019).

| Drugs | Frequency | Quantity Consumed |
|---|---|---|
| Cotrimoxazole 480 mg tablet B/1000 * | 29,301 | 180,492,898 |
| Amoxicillin 250 mg capsule B/1000 | 24,024 | 179,323,559 |
| Paracetamol 500 mg tablet B/1000 | 23,515 | 56,048,800 |
| O.R.S sachet 20.5 g B/100 | 22,634 | 52,648,404 |
| Chlorpheniramine 4 mg tablet B/1000 | 22,112 | 25,753,669 |
| Nevirapine 200 mg tablet B/60 | 22,037 | 7,434,876 |
| Aminophylline 100 mg tablet b/1000 | 19,475 | 4,372,265 |
| Arte 20 mg + Lumef. 120 mg tab (4 × 6) B/30 | 18,306 | 2,561,125 |
| Cromoglycate Disodic Opht solution 2% | 17,191 | 705,430 |
| Iodine Polyvidone 10% solution 200 mL B/1 | 16,434 | 414,398 |

* The last component of drug name indicates the pack size, as an example, for cotrimoxazole B/1000 = Box of 1000 tablets.

These observations are consistent with the World Health Organization's perspective, which prioritizes citizens' access to medicines and stresses nations' responsibility to build a well-functioning health supply chain as a fundamental pillar in national health systems [12]. In reality, the tremendous financial impact of stock outs of essential medicines and other

life-saving supplies cannot be quantified. As a result, minimizing out-of-stock and boosting customer demand prediction accuracy result in appropriate inventory levels aimed at delivering consumer-centric health services [11,18].

### 3.2. Description of Medicines Consumption—Quantity of Consumed Drugs by the Time

According to Figure 1A,B the quantity of the consumed drug was changing by year and month, but there is a high pick in august 2015. In the analysis, we decided to remove it as any other outlier identified which may result from data entry or other circumstances.

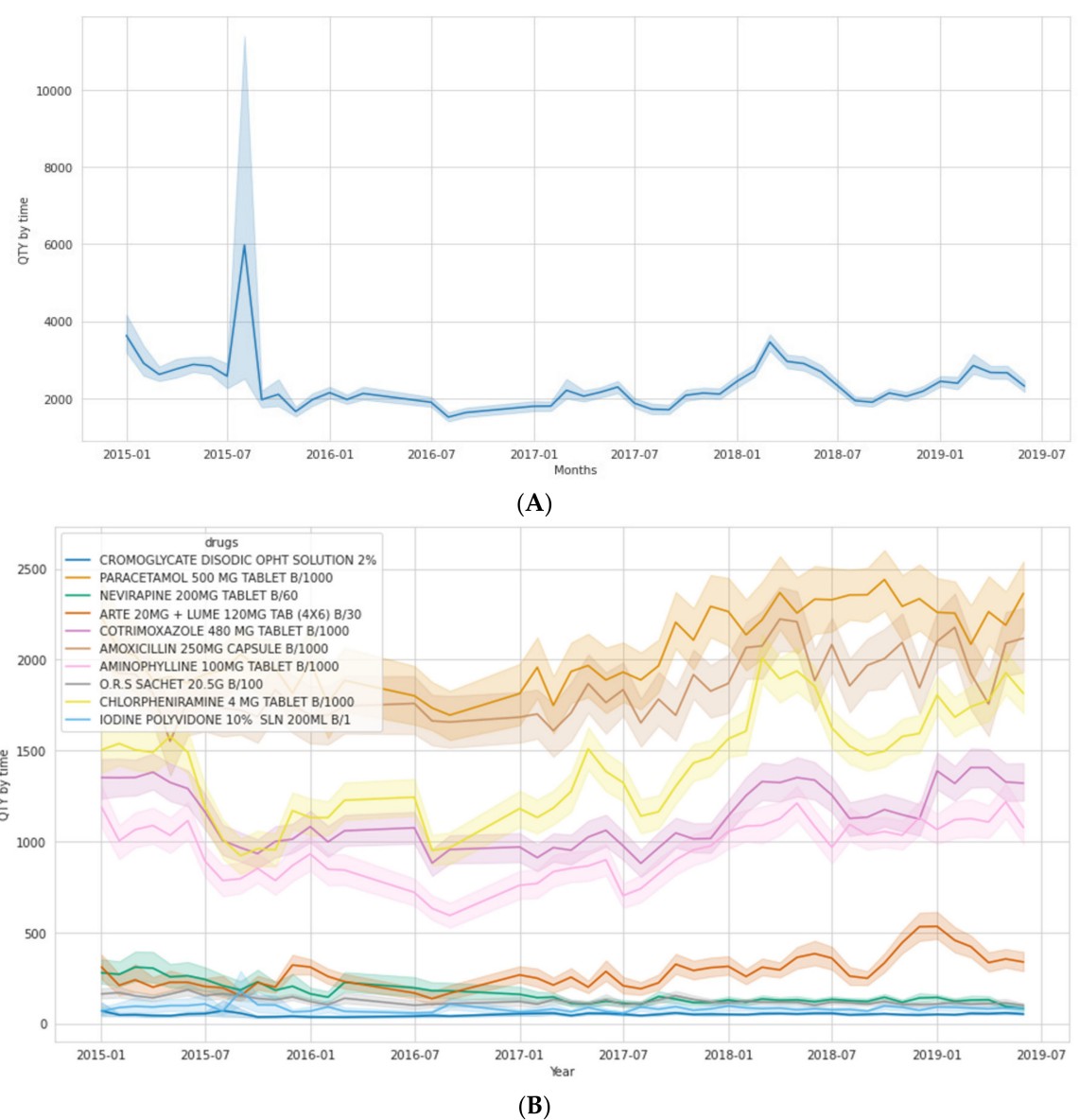

**Figure 1.** (**A**): Quantity of consumed drugs by the time (all selected essential medicines combined). (**B**): Quantity of Consumed drugs by the time (Selected essential medicines independently).

### 3.3. Description of Essential Medicines Consumption by Geographical Distribution

According to the Figure 2, in Rwanda five (5) out of 30 districts were identified to be on top for essential medicines consumptions with a consumption frequency going between 8610–10,162 for ten (10) selected essential medicines. These Districts are namely Nyagatare, Gatsibo, Gicumbi, Gakenke and Karongi.

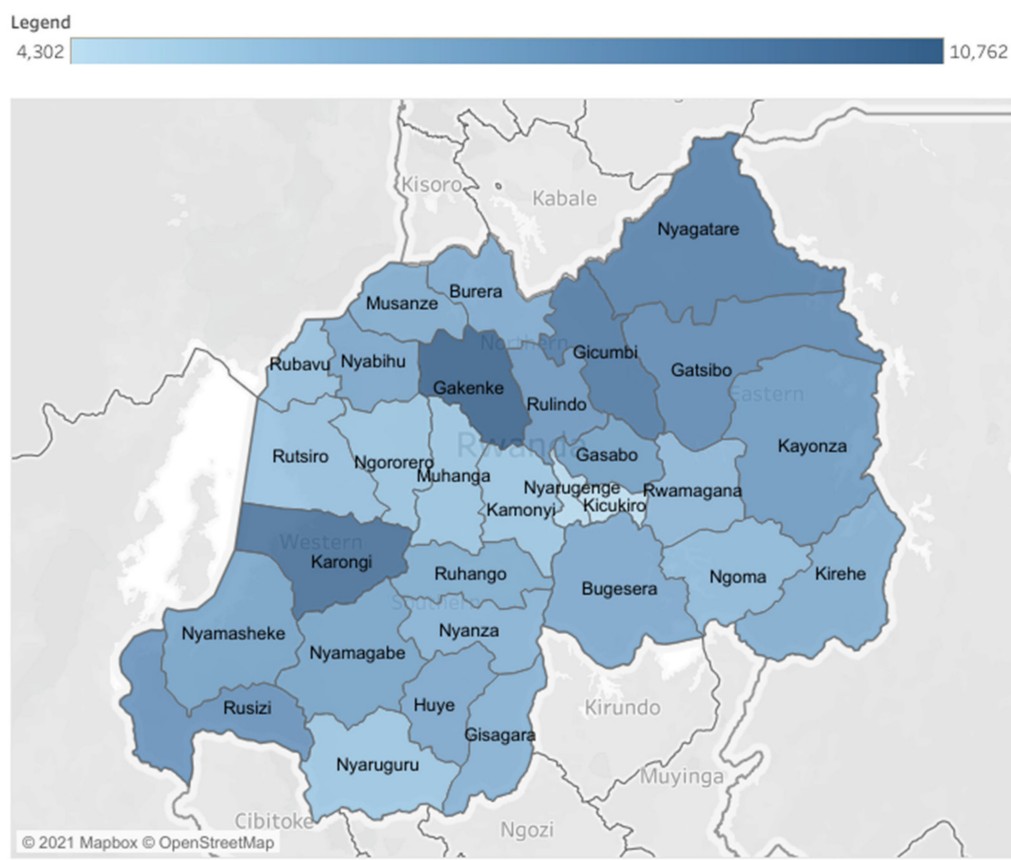

**Figure 2.** Geographical distribution of medicine consumption stratified by district in Rwanda.

As previously stated, the management of the health supply chain for essential medicines is characterized by high complexity and fluctuating demand, as well as supply across multi-level delivery channels that are often strictly regulated and tightly controlled before reaching consumers as end-users [19].

*3.4. Machine Learning Application for Predictive Modelling*

Whilst using ML, we trained the model on one set of observations and test it to another different set of observations to learn the model's generalizability to new data. We have split data into two sets. The train set is data from January 2015 until December 2018, and the test set was from July 2018 until June 2019. We used test data from January to December 2019, but the test data was also required to have a 12-month period as training data to have the same dimensions. For the sake of retaining the enormous training dataset, we included the prior 6 months to balance dimensions without removing it from training. The data were grouped by year and month for better prediction as we want to predict the quantity of the specific drug on basis of a month as indicated in Table 3.

**Table 3.** Summary Statistics of Quantity of drugs consumed for a Train and test set.

| Set | Variable | Count | Mean | Standard Dev. | Median | Min | Max |
|---|---|---|---|---|---|---|---|
| Train | Amount | 149,740 | 834.15 | 1076.45 | 292 | 14 | 4335 |
| | Total amount | 14,321 | 18,871.056 | 24,143.68 | 6515 | 14 | 153,839 |
| Test | Amount | 39,966 | 868.95 | 1110.27 | 281 | 14 | 4335 |
| | Total amount | 3502.00 | 9841.05 | 10,307.60 | 5564.00 | 14.0 | 53,952.00 |

Table 3 presents the summary statistics of quantity of drugs consumed on both train and test set before and after grouping them by year, type of drugs and districts to have data

on monthly basis and by consumption of each district. On train set we had 149,740 observations, with mean of 834.15 of quantity of drugs consumed, standard deviation of 1076.45, median of 292, the minimum value was 14, and maximum value was 4335. For grouped data, the number of observations were 14,321, mean of 18,871.056, standard deviation of 24,143.68, median of 6515, the minimum value equal to 14, and maximum value equal to 153,839. With test set, we had 39,966 observations, with mean of 868.95 of quantity of drugs consumed, standard deviation of 1110.27, median of 281, the minimum value was 14, and maximum value was 4335. For grouped data, the number of observations were 3502.00, mean of 9841.05, standard deviation of 10,307.60, median of 5564.00, the minimum value equal to 14, and maximum value equal to 53,952.00.

### 3.5. Prediction of Essential Medicines Demand with Machine Learning Techniques

Predictive modeling is an approach that use a mathematical method to foresee future occurrences or outcomes, as well as to predict future trends, by searching for patterns that have occurred in the past or by analyzing historical data [20]. In this point of regards, businesses with limited resources, based on experience, can use readily available software to develop reliable and accurate estimates in a timely and cost-effective manner using excel spreadsheets or other relatively simple excel- based software [19]. Even though these are simple procedures with limited insight, it was evidenced that they are indeed widely used with an estimated rate of 82.1% [20]. Indeed, in the field of health supply chain, the purpose of predictive modeling is to answer to the concern regarding the use of known past behavior, for anticipating the most likely scenario to happen in the future [21]. In this study, the linear regression, ANN, and Random Forest models were used to predict future trends for essential medicines.

Linear Regression: Linear regression is a mathematical function and supervised ML model that predicts new data using input data with labels. The model looks for the best-fitting linear line between the response and the explanatory components. It's used to make predictions, particularly when the answer variable is numerical or quantitative [22].

The following is the linear regression equation

$$Y = \beta_0 + \beta_1 X_1 + \beta_2 X_2 + \beta_3 X_3 + \ldots\ldots + \beta_n X_n \tag{1}$$

$\beta_0$ is the intercept, $\beta s$ are coefficient, and $X$ are predictor variables.

Random Forest: Random Forest is an ensemble learning technique for regression and other tasks that builds a huge number of judgmental choices during training. As the name indicates, a Random Forest is a tree-based ensemble in which each tree is determined by a sequence of random variables [23]. In more formal terms, we assume an unknown joint distribution PXY for a p-dimensional random vector $X = (X1\ldots Xp)$ T expressing the real-valued input or forecaster variables and a random variable Y indicating the real-valued response $(X, Y)$. The aim is to explore a prediction task f($X$) for predicting $Y$. The random forest algorithm which has been used is composed of 18,000 number of estimators (n_estimators), 50 max depth (max_depth), 14 maximum features (max, features), 4 minimum sample leaf (min_sample_leaf), and random state (random_state) set to zero [23,24].

Artificial neural network: ANN is a modeling technique based on the human nervous system that allows for learning by example from representative data to explain a real-life event or a process leading to decision—making. As indicated in Figure 3, a defining aspect of ANN is its capacity to build empirical connections between independent and dependent variables, as well as extract nuanced information and sophisticated knowledge from representative data sets. ANN models provide numerous advantages over regression-based models, such as the capacity to manage outliers, because the links between variables may be formed without making any assumptions about a detailed description of the events [22].

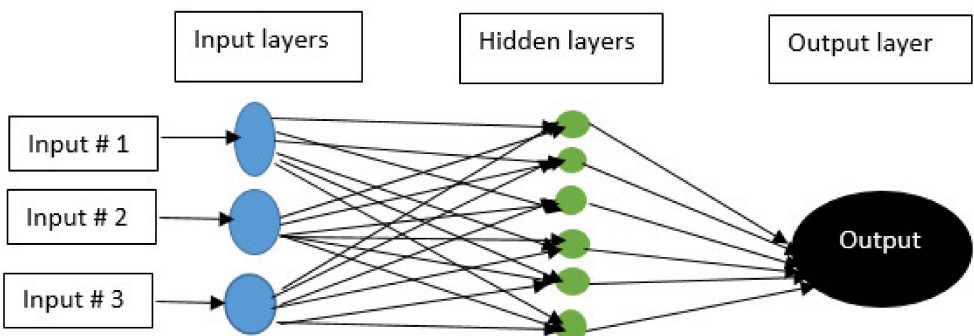

**Figure 3.** The basic structure of artificial neural network (Source: https://data-flair.training/blogs, accessed on 25 October 2021).

The basic structure of ANN used was composed of 11 dense layers, the last one as output, the first dense layers were composed of 300 units, with an input shape of 53 equal to the number of predictors, the second dense layer was composed of 300 units, the third and fourth dense layers were composed of 150 units, the fifth and sixth dense layers had 75 units, the seventh and eighth had 50 units, ninth and tenth had 25 units, the last dense layer which is output layer had only one unit for prediction.

The activation function used is relu, l2 regularizer and was used with a rate of 0.08 to regularize the weight to avoid over-fitting. In our study, we used the Adam optimizer to build the model, with mean squared error as the loss function, mean squared error and accuracy as training metrics, 64 batch size, and 1000 epochs. In actual fact, Forecast Bias, Mean Absolute Deviation (MAD), Mean Squared Error (MSE), and Mean Absolute Percentage Error (MAPE) are the most widely functional demand forecast accuracy metrics to assess the validity of models, according to prior research [8,25]. The accuracy metrics applied, how it is introduced, deployed, and tracked are all influenced by the type of output generated.

Realistic and evidence-based predictive modeling has the potential to provide optimal critical medicines availability while decreasing safety inventories, reducing waste, and continual improvement in the management of essential medicines stock levels and store replenishment [26]. Despite the existence of links between accuracy-based metrics and levels of aggregated prediction across commodities and forecasting periods, as well as the matching of stated accurateness and standards, forecasting errors remain. Of course, forecasting errors are known to vary depending on at which extent forecasters are involved in the supply chain activities. However, forecasts made closer to the demand point will be more accurate. However, forecasts made farther up the management of supply chain by heightening the control of incidental forecast inaccuracies [27].

Model Evaluation: After training, we evaluated the model by comparing its predictions to the true target on training data. However, we also evaluated our model on new data that had not been utilized in training. R-square ($R^2$) and root mean square error (RMSE) were used to evaluate the model.

Root mean square error: Considering that in RMSE, the errors are squared before being averaged. This means that the RMSE weights larger errors more heavily. This implies that when large mistakes are present and have a substantial influence on the model's performance, RMSE is significantly more effective. Along with this feature is beneficial in many mathematical computations since it avoids calculating the absolute value of the mistake. In this metric as well, the lower the value, the better the model's performance [25].

R-square ($R^2$): This metric indicates how well a model matches a given dataset. It's also referred to as the coefficient of determination. The $R^2$ indicates how close a regression line (the displayed predicted values) is to the actual data values. In run through, the $R^2$ value varies from 0 to 1, with 0 indicating that the model does not fit the data and 1 indicating that the model perfectly fits the dataset [25].

## 4. Discussion of Experimental Findings

As presented in Table 4, from our experimental results, Random Forest model is better than other applied models, as it has low RMSE and the model fit 88 percent of the training dataset and 76 percent on test data than Linear regression that fit 80 percent on the training dataset and 74 percent on the test set, as well as 87 percent for the neural network on the training set and 55 percent on test set.

**Table 4.** Presentation of Model Results.

| Model | Set | RMSE | R-Square |
|---|---|---|---|
| Linear Regression | Train | 0.82 | 0.80 |
| | Test | 0.89 | 0.74 |
| Random Forest | Train | 0.64 | 0.88 |
| | Test | 0.84 | 0.76 |
| Neural Network | Train | 3429.98 | 0.87 |
| | Test | 6922.77 | 0.55 |

The figures of prediction models for essential medicines consumed by month versus the actual number of drugs by each model are presented as indicated in the Figures 4–6.

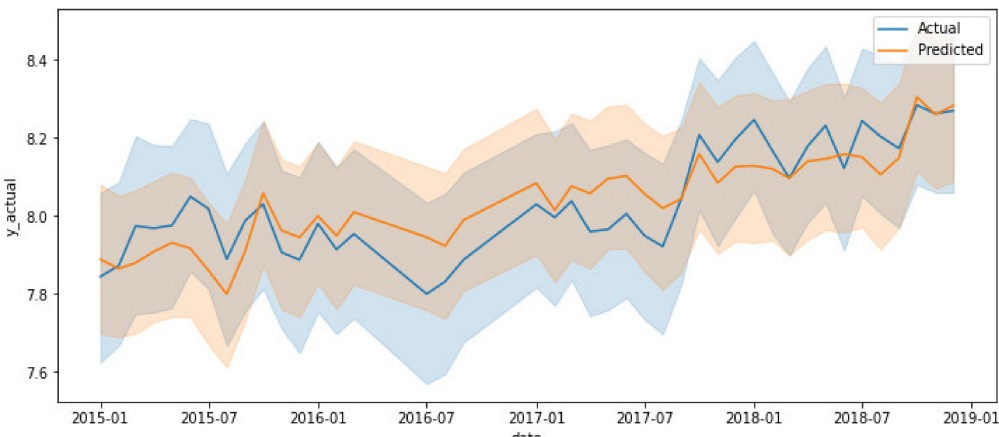

**Figure 4.** Linear Regression.

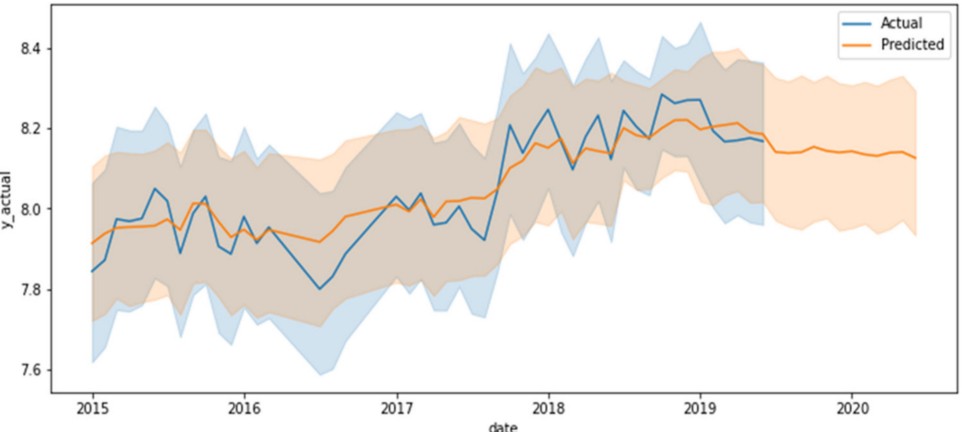

**Figure 5.** Random Forest.

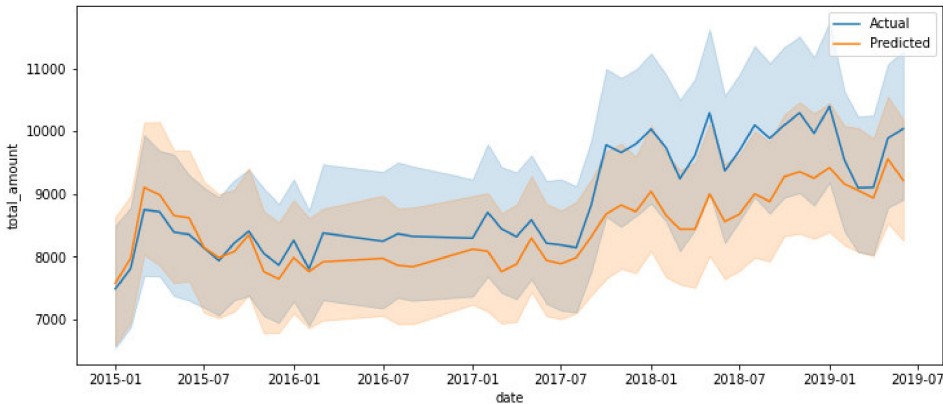

**Figure 6.** Artificial Neural Network.

The Linear Regression model predicted approximately equal to the actual values, and it was able to predict the trend.

The Random Forest model predicted approximately equal to the actual values; it was able to predict the trend very well.

The Artificial neural network model predicted approximately equal to the actual values, it was able to predict the trend but, in the end, the predictions are higher different than actual.

Root mean squared error of Linear regression on train data equal to 0.82, on test data equal to 0.89, for Random Forest is 0.64 on a train set, on a test set is 0.84, and for the neural network is 3429.98 on a train set, on a test set is 6922.77. Neural network model actual numbers were used to predict as the log transformed values were not able converge so we kept the real values in model training. R-square of Linear Regression on train data equal to 0.80, on test data equal to 0.74, for Random Forest is 0.88 on a train set, on a test set is 0.76, and for the neural network is 0.87 on a train set, on a test set is 0.55.

According to our experimental results, Random Forest model is better than others as it has low RMSE and the model fit 88 percent of the training dataset and 76 percent on test data than Linear regression that fit 80 percent on the training dataset and 74 percent on the test set, as well as 87 percent for the neural network on the training set and 55 percent on test set. The higher difference of train and test metrics in that case the model overfitted, all the models have over fitted but neural network overfitted highly.

More complex methods for predicting difficulties that use a system dynamics modeling approach have arisen in recent decades, contributing to the creation of models that track the control and management of diseases and the use of essential medicines over time and form loops in prediction models [28].

According to previous studies, while there is plenteously data that are useful for more precise and reliable demand forecasting (e.g., changes in disease control, treatment guidelines, and their effects on consumption of essential medicines), data related to treatment and medicines prescription is limited due to a variety of factors including different data formats, a lack of management tools for data integration, data collection times and real fact from data, as well as lack of new and advanced models for vastly increased prediction insights [29]. Modelling, for example, is indeed a useful way to verify prediction models since it uses data from the past. Data visualization also aids collaborative decision-making and forecasting future demand for vital drugs in the healthcare supply chain.

In the management of health supply chains and pharmaceutical commodities, time-series models are used the most (52 percent) and causal models are used the least (24 percent), according to a 2002 study by Jain, while judgmental models account for 19 percent and mixed or combination models account for 5% [30]. Furthermore, leveraging AI technology such as ML applications to improve the accuracy in forecasting and demand predictions for essential medicines might potentially improve their availability and reshape their well-ordered distribution [31,32].

## 5. Conclusions and Recommendations

Given that the supply and management of essential medicines is challenging, and as a necessity, there are several obstacles to overcome to maximize their availability and increase access to clients who seek them. Applied predictive modeling on essential medicines consumption form a basis for all planned activities in health supply chain. According to several research, forecasting is a comparatively recent understanding for the healthcare field, which may explain the popularity of basic approaches mostly conducted using excel spreadsheets in the past.

Our study has focused on the application of ML for predicting the future trend for essential medicines consumption in Rwanda. Three types of models have been developed including—Linear regression, Random Forest, Artificial Neural Network models. Two out three developed models were able to fit the data 75 percent at training set, and above 55 percent at test set. Random Forest model outperformed others at 88 percent on train set and at 76 percent on test set.

The random forest was able to predict data accurately at 88 percent with train set and 76 percent with test set and thus it can be used to make predictions for future consumptions based on past consumption by inputting the year, month, districts, and drugs (among 8 used in training). Finally, the findings of our study's experimental examination of three forecasting scenarios demonstrate that the random forest predicting model has the greatest match to historical demand data, reduced error estimates across the look at scenarios and trials.

Given our findings, we strongly recommend that the random forest model be used to predict future trend in demand for essential medicines. The availability of medicines will be ensured and optimized by taking such predictions into account. Furthermore, our study revealed very pertinent information that could have an impact on the effective and efficient management of the health system, as well as improving Rwandans' well-being through the management of a reliable and sustainable supply of essential medicines and other life-saving products. The random forest model has developed a unique method for effectively forecasting future trends in vital drug demand, making it a powerful recommendation in the field of health supply chain.

**Author Contributions:** Corresponding author and Co-Authors has contributed to this study as follow: F.M. and E.K., J.N. (Japhet Niyobuhungiro), J.N. (Joseph Nkurunziza) contributed to study conceptualization; F.M., J.N. (Japhet Niyobuhungiro) and J.N. (Joseph Nkurunziza) contributed to the methodology; F.M. and J.N. (Japhet Niyobuhungiro), Predictive Modelling and ML application; F.M., E.K. and J.N. (Joseph Nkurunziza).; validation, F.M., J.N. (Joseph Nkurunziza), E.K. and J.N. (Japhet Niyobuhungiro).; formal analysis; F.M. contributed to the investigation, F.M.; resources, F.M. and J.N. (Japhet Niyobuhungiro); data curation, F.M.; writing—original draft preparation, F.M.; writing—review and editing, F.M. and J.N. (Japhet Niyobuhungiro); visualization, E.K., J.N. (Joseph Nkurunziza) and J.N. (Japhet Niyobuhungiro); supervision, F.M. and E.K.; project administration, F.M. and ACEDS; funding acquisition. All authors have read and agreed to the published version of the manuscript.

**Funding:** This research received no external financial funding, but the research has been conducted under an academic support offered by the University of Rwanda through the African Center of Excellence in Data Science (ACE-DS). Article processing charge will be supported by the ACE-DS.

**Institutional Review Board Statement:** The IRB statement is not applicable. The study used routine program data extracted from eLMIS, which is a digital tool used in management of medicines in Rwanda. The National Health Research Committee authorized the collection of data through the approval with Ref No: NHRC/2020/PROT/015, and the Ministry of Health (Rwanda) provided a research collaborative note and authorized the access to data.

**Informed Consent Statement:** Not applicable. The study used routine program data and did not involve humans or animals.

**Data Availability Statement:** The study made use of historical data from Rwanda's health supply chain and related datasets such as eLMIS. The Ministry of Health has approved the use of eLMIS data.

**Acknowledgments:** This research was conducted as part of a Research Project entitled "An Application of ML in Digital Supply Chain for Optimizing the Availability of Essential Medicines in Rwanda "at University of Rwanda through the African Center of Excellence in Data Science (ACE-DS). The ACEDS is one of 24 African Centers of Excellence supported by the World Bank. It has been selected and established at the University of Rwanda in College of Business and Economics on 17 October 2016. It is a regional center, and it combines expertise in statistics, economics, business, computer science, and engineering to use big data and data analytics to solve development challenges.

**Conflicts of Interest:** The authors declare no conflict of interest. The funders had no role in the design of the study; in the collection, analyses, or interpretation of data; in the writing of the manuscript, or in the decision to publish the results.

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
