# Peer review of "The Prediction of Essential Medicines Demand: A Machine Learning Approach Using Consumption Data in Rwanda"

_processes, doi:10.3390/pr10010026_

Round 1

Reviewer 1 Report

This is an interesting manuscript addressing the prediction of essential medicines used in Rwanda. I believe Authors need to address some points written below before publication.

Methods

L136-138 Why did you employ these five variables? It seems that one of them (the amount of drug consumed in quantity) is dependent variable, and the name of drug consumed is just a label. Therefore, you predicted the amount of drug consumed in quantity with three independent variables, namely, districts, health center or hospital, and time, right? I’m also interested in how the districts worked as variable. The district might just represent the population there.

What is B/1000, B/100, or so on? Please explain.

Results

L163 Possibly, the number in a parenthesis “(515)” may be “(23515)”.

L166-167 I cannot understand how you calculated the numbers 179,795,584 and 179,072,549 from presented numbers. I believe it’s better to explain how. This could be applied to the numbers in the right most column.

There is no legends with figures. You may want to put legends with them. 

L196 How did you assess or decide August 2015 as outlier? Did you predefined any threshold? Did you find any reason of the outlier?

L 132-133, L204-207 Authors replaced the name of the district into codes for the confidentiality purpose, but it seems to be opened in Fig 2. I’m skeptical whether the concealment worked in this research.

L215-216 Why did you use data from July to December 2018 as both training and test data? It sounds unusual. 

English

Abbreviations should be spelled out only where they appear first time. In the current version, AI and eLMIS seem to be inappropriately spelled out or abbreviated.

Author Response

Thank you for your constructive feedback - All comments provided and questions asked have served me to improve my manuscript. 

I wish you the best

Reviewer 2 Report

The authors' research was focused on the development of a model to assertive predict the demand for essential medicines in Rwanda. 
 This study is important not only to facilitate the health supply chain planning and operational management but also to ensure that all the population has access to medicines.
Although the study is worth publishing in this journal, the authors should revise some parts.

Introduction 

The introduction is too large and sometimes confusing. The description of the employed program should be described in the materials and methods section.
Line 40 
“After all, the healthcare system is going to undergo a huge data revolution, with AI, predictive….” 
Please define AI

2.1 setting
Please clarify the setting

2.2. Sources, types of data, and data pre-processing
 Please classify the Drugs using the Anatomical Therapeutic Chemical Classification System from WHO.

The selection of the most used drugs was made through the analysis of the number of prescribed medicines or based on the DDD of the prescribed substances?

To better understand the reality of Rwanda regarding the supply and consumption of medicines and to better understand the trends of consumption data should be presented in DDD per patient. 

Conclusion
“Thus, the study concluded that the random forest model has the capacity to fittingly predict the future trend for essential medicines demand”

This work contains very relevant information, however, can you please explain the relevance of the data in terms of the wellbeing of the Rwanda population and the sustainability of the Rwanda healthcare system.

Author Response

Thank you very much for your time to review my paper. I have learned a lot from your comments and suggestions.

Best regards.

Round 2

Reviewer 1 Report

The authors responded to my comments appropriately. Thanks for this opportunity.